

# Particulate emissions from large North American wildfires estimated using a new top-down method

T. Nikonovas, P. R. J. North, and S. H. Doerr

Geography Department, College of Science, Swansea University, Singleton Park, Swansea, SA2 8PP, UK

*Correspondence to:* T. Nikonovas (tadas.nik@gmail.com)

**Abstract.** Particulate matter emissions from wildfires affect climate, weather and air quality. However, existing global and regional aerosol emission estimates differ by a factor of up to 4 between different methods. Using a novel approach, we estimate daily total particular matter (TPM) emissions from large wildfires in North American boreal and temperate regions. Moderate Resolution Imaging Spectroradiometer (MODIS) fire location and aerosol optical thickness (AOT) datasets are coupled with

HYSPLIT atmospheric dispersion simulations, attributing identified smoke plumes to sources. Unlike previous approaches, the method (i) combines information from both satellite and AERONET observations to take into account aerosol water uptake and plume specific mass extinction efficiency in converting smoke AOT to TPM, and (ii) does not depend on instantaneous emission rates observed during individual satellite overpasses, which do not sample night-time emissions. The method also allows multiple independent estimates for the same emission period from imagery taken on consecutive days.

Repeated fire-emitted AOT estimates for the same emission period over two to three days of plume evolution show increases in plume optical thickness by approximately $10\%$ for boreal events, and by $40\%$ for temperate emissions. Inferred median water volume fractions for aged boreal and temperate smoke observations are 0.15 and 0.47 respectively, indicating that the increased AOT is partly explained by aerosol water uptake. TPM emission estimates for boreal events, which predominantly burn during daytime, agree closely with bottom-up Global Fire Emission Database (GFEDv4) and Global Fire Assimilation

System (GFASv1.0) inventories, but are lower by approximately $30\%$ compared to Quick Fire Emission Dataset (QFEDv2) $PM_{2.5}$, and are higher by approximately a factor of 2 compared to Fire Energetics and Emissions Research (FEERv1) TPM estimates. The discrepancies are larger for temperate fires, which are characterised by lower median FRP values and more significant night-time combustion. The TPM estimates for this study for the biome are lower than QFED $PM_{2.5}$ by $35\%$, and are larger by factors of 2.4, 3.2 and 4 compared with FEER, GFED and GFAS inventories respectively. Large underestimation

of TPM emission by bottom-up GFED and GFAS indicates low biases in emission factors or consumed biomass estimates for temperate fires.

## 1 Introduction

Large and often severe fires in boreal and temperate forest regions alter atmospheric composition, considerably affecting the Earth's radiative budget (Langmann et al., 2009; Bond et al., 2013) and degrading air quality (Johnston et al., 2012).

Burning regime in these regions is dominated by episodic extreme events (Stocks et al., 2002) emitting continental scale


plumes (Colarco et al., 2004) with inter-hemispheric transport potential (Damoah et al., 2004; Dahlkötter et al., 2013). Future climate predictions indicate both dryer conditions and greater than average warming for northern latitudes, projecting a likely increase in area burned (Liu et al., 2010) and soil carbon consumption (Turetsky et al., 2015). For the quantification of smoke radiative forcing and impacts on human health, a realistic representation of biomass burning emissions in climate and air quality

models is needed. Disagreement between bottom-up and top-down emission estimates of particulate matter, however, remains large (Kaiser et al., 2012; Ichoku and Ellison, 2014).

Bottom-up emission inventories use emission factors (EF) (Andreae and Merlet, 2001; Janhäll et al., 2010; Akagi et al., 2011; Urbanski, 2014), ratios of gases and particulate matter emitted per unit of dry fuel burned, compiled for different biomes from a range of burning experiment measurements across the globe. Emission factors are applied to biomass burned estimates

which are typically based on satellite observations of ubiquitous but highly variable fire activity. The Global Fire Emission Database (GFED) (van der Werf et al., 2010) makes use of satellite burned area products (Randerson et al., 2012; Giglio et al., 2013) and active fire pixel counts, while the Global Fire Assimilation System (GFAS) (Kaiser et al., 2012) employs fire radiative power measurements (Giglio et al., 2006). Burned area estimates are converted to biomass burned using modelled carbon pools and soil moisture dependent combustion completeness characteristic to the fuel types. FRP based methods rely

on observed relationships between observed FRP and biomass combustion rates (Kaufman and Tanre, 1998; Wooster et al., 2003, 2005).

The more top-down methods utilize satellite aerosol optical thickness (AOT) observations. The Quick Fire Emission Database (QFED) uses regional AOT measurements to scale emissions based on EFs (Darmenov and da Silva, 2015). Similarly, atmospheric model assimilation of GFAS emissions (Kaiser et al., 2012) suggested a 3.4 global correction factor was needed to

reconcile TPM estimates with observed AOTs. Purely top-down methods estimate emissions through inverse modelling of satellite AOT retrievals (Ichoku and Kaufman, 2005; Dubovik et al., 2008). A top-down global gridded Fire Energetics and Emissions Research (FEERv1) (Ichoku and Ellison, 2014) emission coefficients product is based on collocated satellite FRP and AOT observations. The product allows direct conversion from time integrated FRP to emitted particulate matter without invoking the emissions factors.

Global and regional particulate matter estimates from the bottom-up burned area and fire pixel count based GFED agree well with the FRP based GFAS estimates. Model assimilation of these bottom-up emissions, however, suggest TPM underestimation by a factor of 2 to 4, compared to satellite AOT observations (Kaiser et al., 2012). Enhanced GFAS TPM estimates and scaled QFED agree better with top-down FEER emission coefficients on global scales. Notable discrepancies, however, are present for individual regions. North American emissions are larger for enhanced GFAS TPM and QFED when compared to top-down

FEER, while FEER agrees closely with the bottom-up GFED inventory.

A number of uncertainties in both bottom-up and top-down estimates can contribute towards the apparent TPM discrepancies. Average EFs for different biomes conceal the lack of spatial and temporal representativeness for some areas, and large variability in individual measurements introduced by within-biome inconsistencies in vegetation density, climatic and burning conditions (van Leeuwen and van der Werf, 2011; van Leeuwen et al., 2014). Consumed biomass estimates inherit errors of

satellite burned area (Randerson et al., 2012), fire location or FRP retrieval (Giglio et al., 2006), and depend on a range of





assumptions on availability and consumption of carbon in aboveground and soil pools (French et al., 2004). Top-down aerosol inversions are affected by AOT retrieval error and large uncertainties in assumed smoke particle properties, which are required to relate aerosol extinction to particulate mass (Reid et al., 2005b). Moreover, estimates of emission rates based on near source retrievals are representative of burning conditions at the time of satellite overpass. A recent study indicated that night-time

TPM emissions might be underestimated by a factor of 20 - 30 for a large temperate forest fire in Western USA (Saide et al., 2015), stressing the need for better representation of night-time emissions in the inventories. Methods based on regional AOT observations, on the other hand, must take into account poorly constrained ageing effects (Reid and Hobbs, 1998; O'Neill et al., 2002).

This study presents estimates of particulate matter emissions from large wildfires with identifiable plumes in North American

boreal and temperate regions. A newly developed top-down method is applied which attributes satellite aerosol observations to a specific fire event and emission period. Quantified daily fire-emitted AOT takes into account aerosols injected throughout the diurnal cycle and does not rely on instantaneous emission rates observed during a satellite overpass. In some cases, AOT attribution for the same emission period is achieved from satellite images taken on successive days, allowing assessment of uncertainty and investigation of systematic changes in plume optical thickness over time. Total particulate matter is quantified

by applying mass extinction efficiency which is simulated using AERONET particle properties, and accounting for inferred water uptake by aerosols. The results are compared with existing estimates in order to investigate systematic differences between the approaches.

## 2   Data and methods

Daily total particulate matter emissions for large and persistent fire events were estimated by combining Moderate Resolution

Imaging Spectroradiometer (MODIS) active fire observations and aerosol optical thickness retrievals with plume dispersion simulated using the Hybrid Single-Particle Lagrangian Integrated (HYSPLIT) model.

### 2.1   Active fires

To represent fire activity we used the active fire location dataset MCD14ML produced by the University of Maryland and provided by NASA Fire Information for Resource Management System. The data product is based on MODIS mid-range

and thermal infrared observations. MODIS sensors are flown on board the sun-synchronous polar-orbiting Terra and Aqua satellites passing the equator at 10:30 and 13.30 local time during the daytime hours, and 22.30 and 1.30 at night respectively. The instruments have a wide swath of approximately 2330 km, each providing nearly global coverage daily. For high latitudes the coverage is better due to increasing overlap between consecutive overpasses. Each detection in the dataset represents an active fire in a $1\,\mathrm{km}^2$ pixel at the time of satellite overpass, and contains information on the retrieved fire radiative power.



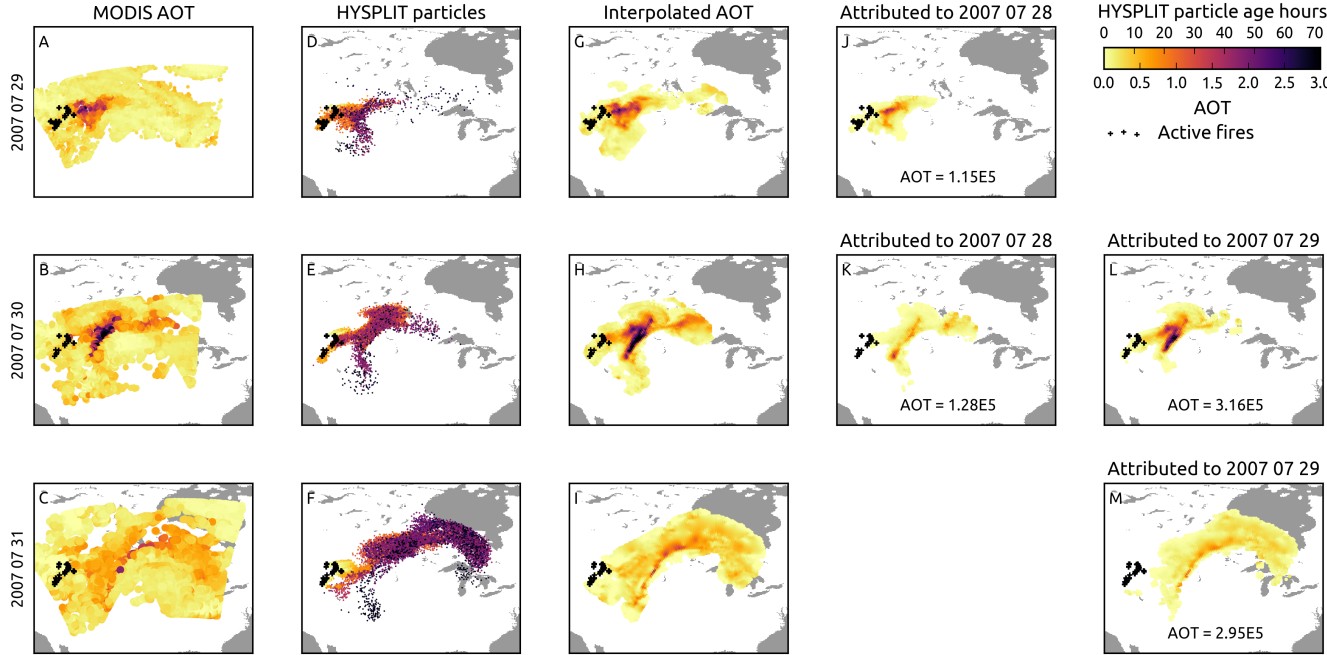

**Figure 1.** An illustration of the method showing an example of fire-emitted AOT attribution for two diurnal cycles of a temperate fire. Rows in the figure represent three successive days of satellite imagery from which the attribution was achieved. Columns from left to right show MODIS AOT retrievals for the day from a single platform with the highest coverage **(A–C)**, snapshots of HYSPLIT particle positions and age taken at local noon **(D–F)**, and AOT interpolated to 25 km equal area grid **(G–I)**. The two right columns show fire-emitted AOT attributed to 28[th] **(J)** and **(K)** and 29[th] **(L)** and **(M)** of July 2007 determined from images taken on different days. Total attributed AOT is shown within the plots.

## 2.2 Fire event selection

The active fire identifications were agglomerated into large wildfire events by grouping any pixels located closer than 150 km in space and 24 hours in time. For the analysis we selected events larger than $100 \, \text{km}^2$ and with duration longer than 7 days, as they were likely to be strong emission sources. Fires of such size or larger are a dominant mode of burning in North American boreal and temperate forests contributing more than 80% to total burned area in these regions (Stocks et al., 2002; Kasischke et al., 2002). Burning events were classified into boreal and temperate fires using the dominant emission source given in the GFEDv4 inventory for areas and periods when the events were active.

## 2.3 Plume dispersion modelling

Smoke dispersion for the selected fire events was simulated with the HYSPLIT model (Draxler and Rolph, 2003). The model was run using Global Data Assimilation System (GDAS) meteorological archive data. For each day of burning particles were





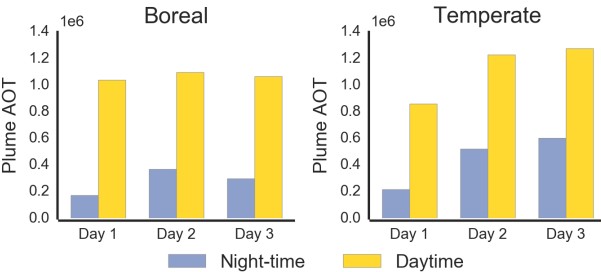

**Figure 2.** Changes in attributed AOT over time. Shown are 39 boreal and 37 temperate diurnal emission cycles for which estimates were obtained on three consecutive days, for both daytime and night-time periods.

continuously injected and vertically distributed within the planetary boundary layer. Particle number was scaled proportional to active fire pixel counts observed during the satellite overpasses. For each diurnal cycle when the fire event was active, particles were injected at two different rates for two 12 hour intervals representing day and night emissions 09:00 to 21:00 and 21:00 to 09:00 local time respectively. The injection rates were scaled in proportion to the highest detected fire pixel count from either Aqua or Terra overpasses during the time periods. When no fires where detected the count was set to a minimum positive value estimated for the fire event from all daytime or night-time observations. Modelled particle positions at local solar noon were used to compare against Terra and Aqua Aerosol Optical Thickness (AOT) observations.

## 2.4 Satellite aerosol data

MODIS AOT collection 5.1 data products M*D04_L2 were used in this study. The dark target algorithm (Kaufman and Tanre, 1998; Levy et al., 2009) retrieves AOT at 550nm and $10\,\text{km} \times 10\,\text{km}$ spatial resolution at nadir. Pixel size increases with view angle and is about twice the size at swath edges. The AOT product validation against ground-based AERONET AOT observations suggest a one sigma error which increases linearly with aerosol loading $\pm(0.05 + 0.15\,\%)$ (Levy et al., 2010). The AOT retrieval values have upper limit of 5.0, and in addition, opaque smoke is often rejected as bright surface or cloud by the algorithm (Livingston et al., 2014), preventing retrievals over extremely optically dense plumes. Consequently, AOT near the emission source is often not retrieved and the algorithm performs better when plumes are dispersed into regional haze.

## 2.5 AOT attribution

Elevated MODIS AOT observations were attributed to a specific fire event and emission period by comparing regional retrievals to particle positions modelled by HYSPLIT (Fig. 1). For each day of fire activity, MODIS AOT observations from either Aqua and Terra platforms with the highest coverage for the day were matched with modelled plume extent. The matching was performed iteratively for the modelled plume regions dominated by particles emitted during the previous 1 to 6 twelve hour emission periods, representing three full diurnal cycles. AOT attribution was performed for the plume regions with (i) at least 80 % of area with available AOT retrievals from either Aqua or Terra platforms, assuming that a single MODIS AOT pixel





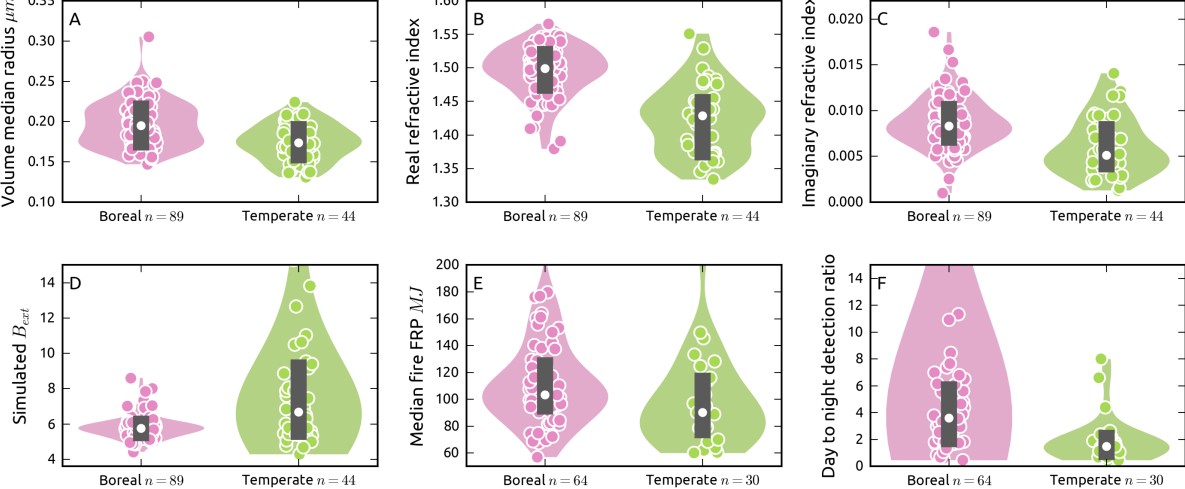

**Figure 3.** Distributions of AERONET-retrieved particle properties **(A–C)** attributed to boreal and temperate fires, simulated mass extinction efficiencies **(D)**, median MODIS FRP values **(E)** and ratios of daytime to night-time active fire detection counts observed during a single overpass **(F)**. Shown are kernel density estimates and individual observations; boxes indicate median values and interquartile range.

represents $100 \, \text{km}^2$ area, and (ii) with AOT median value higher than the estimated background value for the fire event. The background value for a fire event was determined from the median value of the AOT retrievals in the fire region observed two days prior to ignition. MODIS AOTs for the selected cases were interpolated to a $25 \, \text{km}$ resolution equal area grid employing radial basis function interpolation with a linear kernel. The background value was subtracted from within-plume AOTs and

5   negative values set to zero. AOT retrievals above the background value were attributed to different emission periods and different sources. The attribution was performed by partitioning a grid cell's fire-emitted AOT in proportion to the numbers of modelled particles emitted during the emission periods and with origin found within the cell.

## 2.6 Smoke aerosol properties

The Aerosol Robotic Network (AERONET) (Holben et al., 1998) level 2 retrievals (Dubovik and King, 2000) of aerosol mi-

10   crophysical and optical properties were used to characterise particles in plumes under investigation. AERONET consists of ground-based globally distributed sun-sky scanning photometers with a narrow field of view. The instruments are continuously monitored and calibrated, and the retrieved properties have estimated accuracy ranges. The direct sun beam extinction measurements provide spectral AOT at several wavelengths ranging from 0.34 to $1.02 \, \mu\text{m}$ with uncertainty of 0.01 - 0.02 (Dubovik et al., 2000). Measured AOT and angular distribution of sky radiances are used to retrieve column integrated aerosol volume

15   size distribution at 22 size bins from 0.05 to $15 \, \mu\text{m}$ and spectral refractive index at 0.44, 0.67, 0.87 and $1.02 \, \mu\text{m}$. Size retrieval is expected to be accurate within $25 \, \%$ for particles with radii between 0.1 and $7 \, \mu\text{m}$ and $25–100 \, \%$ for size bins outside this





**Table 1.** Real ($n$) and imaginary ($k$) parts of refractive index, and density ($p$) of the components used in the Maxwell-Garnett effective medium approximation calculations. All components were assumed to have spectrally flat refractive index. Uncertainty in $p$ for the species represented by the second inclusion was propagated into combined errors of retrieved water volume fraction and particle density.

| Species | $n$ | $k$ | $p \, \mathrm{g/cm^3}$ | Source |
|---|---|---|---|---|
| Black carbon | 1.95 | 0.79 | 1.8 | Bond and Bergstrom (2006) |
| Organic and inorganic compounds | 1.53 | 0.00 | 1.2–1.4 | Kirchstetter et al., (2004), Turpin and Lim (2001), Toon et al., (1976) |
| Water | 1.33 | 0.00 | 1.0 | |

range. Scans at high aerosol loadings (AOT $0.44 \, \mu m \geq 0.4$) allow retrieval of refractive index with estimated uncertainties of 0.04 and 30 % for real and imaginary parts respectively (Dubovik et al., 2000).

Available observations within areas identified by the dispersion analysis as biomass burning plumes were attributed to a specific emission event and land cover type. Only retrievals containing refractive index (AOT $0.44 \, \mu m \geq 0.4$) were selected. In order to minimize the presence of dust and urban aerosol dominated retrievals, cases with volume concentration of fine mode (particle diameter $< 1 \, \mu m$) fraction less than 0.8, sphericity parameter lower than 0.98 and absorption Ångström exponent lower than 1 were filtered out. To make the samples more representative of plumes for which particulate matter was estimated, we selected AERONET observations within-plume areas dominated by particles aged for 1 to 3 days.

## 2.7 Water content retrieval

The available AERONET spectral refractive indices were used to infer smoke aerosol water uptake. We employed the Maxwell-Garnett effective medium approximation (Bohren and Huffman, 1983) which provides a method to derive volume fractions of the components in the mixture if their refractive indices are known. The approach is described in detail and demonstrated by Schuster et al. (2005) for retrieving black carbon concentrations from AERONET climatologies. It was further developed to infer brown carbon content (Arola et al., 2011), aerosol water uptake (Schuster et al., 2009), and to simultaneously retrieve fractions of carbonaceous absorbers and dust (Schuster et al., 2016).

To infer water content we employed a three component mixture of black carbon and organic-inorganic matter inclusions in water host (table 1). For black carbon we assumed the refractive index and density suggested in Bond and Bergstrom (2006). The second inclusion was used to represent a broad range of chemical species observed in biomass burning plumes (Brock et al., 2011), including organic carbon, organic matter, ammonium sulphate and ammonium nitrate. These species were represented by a single component because they have $n$ values close to 1.53. This value is characteristic to dry ammonium sulphate (Toon et al., 1976), was measured for organic carbon (Kirchstetter et al., 2004) and lies within the range of values measured for dry organic compounds (Dick et al., 2007). Volume fractions of the inclusions and water host were retrieved in two steps. First, we deduced black carbon utilising spectral imaginary refractive index of the component. The Maxwell-Garnett mixing rule was applied for a range of different fractions of black carbon in water host with negligible imaginary index. Volume fraction of the





inclusion was estimated determining the configuration which provided minimum $\chi^2$:

$$\chi^2 = \sum_{i=1}^{N} \frac{(k_i^{ret} - k_i^{mg})^2}{(k_i^{ret})^2}, \tag{1}$$

where $k_i^{ret}$ is AERONET-retrieved imaginary index, $k_i^{mg}$ is the value calculated by the Maxwell-Garnett mixing rule, $i$ is summation over the selected AERONET wavelengths. We used AERONET $k$ at 0.87 and 1.02 μm to retrieve black carbon

fraction, assuming that it is the only absorber at this part of the spectrum. $k$ at shorter wavelengths can be enhanced by absorption by organic carbon (Kirchstetter et al., 2004), which is retrieved as a part of the second inclusion. After volume fraction of black carbon was established, we kept it fixed and varied the fraction of the second inclusion in the mixture, minimizing the equation (1) for real part of the refractive index at all four AERONET wavelengths.

## 2.8 Conversion of aerosol optical thickness to mass

Particle mass within the atmospheric column can be inferred from smoke AOT observations if mass extinction efficiency ($B_{ext}$) is known:

$$M_{plume} = \frac{\tau_{plume}}{B_{ext}}, \tag{2}$$

where $M_{plume}$ is mass of plume aerosols, and $\tau_{plume}$ is a product of mean fire-emitted AOT and plume area. $B_{ext}$ represents extinction in area units per unit of aerosol mass, usually expressed as $[m^2/g]$. It can be measured or calculated invoking Mie

theory. In-situ measurements of fresh North American smoke suggest $B_{ext}$ values ranging from 3.9 to 4.6 m$^2$/g (Hobbs et al., 1996). Equivalent measurements for aged plumes are not available for the region, but smoke samples collected in other forest ecosystems indicate slightly larger $B_{ext}$ values ranging 4.0 to 5.3 m$^2$/g (Reid et al., 2005b; Chand et al., 2006) for older emissions. Similar $B_{ext}$ at 550nm ranging from 4.5 to 5.2 m$^2$/g were inferred by Reid et al. (2005b) from AERONET retrievals (Dubovik et al., 2002) of dominant particle size distributions and index of refraction for North American boreal regions. (Ichoku

and Ellison, 2014) applied a uniform 4.6 m$^2$/g value (Reid et al., 2005b) in deriving FEER TPM emission coefficients. Notably, plumes in their analysis were relatively young, up to a few hours old at most. In contrast, smoke discussed in this study is aged for few days.

To avoid making assumptions on smoke optical properties, $B_{ext}$ was inferred utilizing available AERONET-retrieved refractive indices and particle size distributions. We used Mie code (Bohren and Huffman, 1983) to calculate $B_{ext}$ assuming spherical

internally mixed particles:

$$B_{ext} = \frac{\int_{r_{min}}^{r_{max}} \sigma_{ext}(n,k,\lambda,r) \frac{dN(r)}{d\ln r} d\ln r}{V_{dry} \, p_{dry} \, \frac{3}{4}\pi \int_{r_{min}}^{r_{max}} r^3 \frac{dN(r)}{d\ln r} d\ln r}, \tag{3}$$

where $\sigma_{ext}$ is the extinction cross section of a single particle which depends on refractive indices $(n,k)$, wavelength and particle radius $(r)$. $V_{dry}$ is particle dry volume fraction, $p_{dry}$ is particle dry fraction density, both determined from aerosol water uptake





analysis (section 2.7). $\sigma_{ext}$ was calculated at 0.55 μm using Mie code for every radius in the AERONET size distribution and averaged $n$ and $k$ retrievals at 0.44 and 0.67 μm. The numerator in the equation 3 is single particle extinction cross sections integrated over number distribution, while denominator is aerosol dry fraction mass within the column given by the product of particle density and integrated particle volume.

## 2.9 Uncertainty in derived quantities

Uncertainties in AERONET smoke aerosol properties, particle density and daily fire-emitted AOT attribution were propagated using a Monte Carlo method retrieving water volume fraction, mass extinction efficiency and deriving total TPM estimates for the biomes. Throughout the study we report median values and interquartile range for the distributions, unless otherwise stated.

## 3 Results and interpretation

Attribution of fire-emitted AOT for at least two diurnal cycles of emission was achieved for 94 large fire events. Boreal sources constitute 64 of the events, with the remaining identified as temperate forest fires. In total, fire-emitted AOT estimates were obtained for 620 days of burning. The daily attributed AOT include particulate matter emitted during the full diurnal cycle of emission accounting for both daytime and night-time emissions. These estimates are representative of large and likely intense burning events and clear sky conditions for which sufficient satellite observations were available. Particulate matter emitted by the events on the days for which our estimates were obtained account for approximately 3 to 20 % of total GFED and GFAS emissions for the North America region depending on the year. The representativeness, however, is probably better than suggested by this figure, assuming that emissions from the sampled events were similar on the days for which estimation was not achieved.

### 3.1 Systematic changes in plume attributed AOT

An important advantage of the AOT attribution method presented in this study is that it allows us to gauge combined error originating from uncertainties in plume injection height, dispersion modelling, MODIS AOT retrievals and applied interpolation. Critically, any systematic changes in fire-emitted smoke optical thickness in evolving plumes can be inferred as well. This was facilitated by a number of cases when two or more AOT attributions based on imagery taken on consecutive days were performed for the same emission period. Figure 2 shows daily AOT estimates for days of emission for which the attribution was achieved from imagery taken on three consecutive days, for both night-time and daytime emission periods.

Overall, determined smoke AOT based on retrievals at later stages of plume development tend to have a positive bias compared to estimates for the same period of emission obtained on previous days. Notably, the largest increase in estimated AOT is observed when comparing estimates for the previous night-time emission cycle (smoke aged for 3 to 15 hours) to AOT attributed to the same period determined from the following day's imagery, after the plume has aged for an additional 24 hours. Inferred changes in daytime fire-emitted AOT over the first two days of ageing are smaller. Optical thickness for temperate smoke increases by approximately 30 % from the first observation of daytime emissions which are already aged





for 15 to 27 hours, compared to estimates for the same emission period determined from the imagery collected the following day. Changes in estimated daytime fire-emitted AOT for boreal plumes appears to be negligible. Notably, consecutive 24 hours of ageing does not change estimated plume AOT significantly for both biomes and both daytime and night-time emissions. A slight decrease in optical thickness is observed for boreal smoke, but this should be treated with caution given the level of

uncertainties involved. For the limited number of emission cycles presented in figure 2, contributions of day and night emissions appear to differ between the biomes. Night-time emissions constitute 30–40 % of total fire-emitted AOT for temperate events. Boreal plumes are dominated by daytime emissions with night-time emissions comprising under 20 % of total daily AOT. The difference is influenced by generally larger number of night-time active fire pixels observed for temperate fires (Fig. 3 (F)) and, consequently, more particles released during night-time emission period in the dispersion simulations.

The effect of increasing AOT over time could be in part explained by uncertainty in plume dispersion modelling. However, the modelling error is expected to increase with time and hence should be manifested by progressively larger disagreement and biases for older estimates. In contrast, the results suggest that the agreement between two estimates for the same emission period is smaller at later stages of plume development. The bias, on the other hand, is clearly largest for the first and the second plume observations within the first two diurnal cycles. It is possible that the model-emitted night-time particles get mixed with

subsequent daytime emissions during the transport, effectively scavenging part of AOT from the other emission periods during the attribution. However, the observed daytime AOT tends to increase as well. Additionally, there are significant differences in inferred AOT changes between boreal and temperate plumes, indicating that some physical processes might be driving the change.

Particulate matter estimation and comparison with other methods are based on fire-emitted AOT during emission cycles

starting and ending at 00.00 UTC. For 159 and 125 emission periods for boreal and temperate events respectively, AOT was determined from imagery taken on consecutive days allowing us to estimate the attribution error. These estimates do not include the problematic previous night emissions. Figure 5 shows the differences in fire-emitted AOT estimates for these cases. Given that the differences are approximately normally distributed, we propagated 50 % one sigma uncertainty in attributed daily fire-emitted AOT to derive confidence intervals for TPM emission estimates.

## 3.2  Fire FRP and daytime - night-time pixel counts

Large and persistent fire events discussed in this study exhibit distinctiveness in fire radiative power (FRP) values and diurnal burning cycle. Median MODIS FRP retrieved for the boreal fires is 103 (94–117), while median FRP for temperate events is 90 (78–103) MW. This suggests higher burning intensity and combustion rates for boreal fires. A more striking difference, however, emerges when comparing ratios of maximum active fire pixel counts detected during individual daytime and night-

time satellite overpasses. The proportion of active fires at night are typically much higher for temperate fires. The average daytime to night-time pixel count ratio is 1.4 (1.1–1.9) for the fires in this biome compared to median value of 3.6 (1.8–4.8) for boreal fires. Such a pattern indicates a higher contribution of night burning for temperate events and potentially more important smouldering combustion.





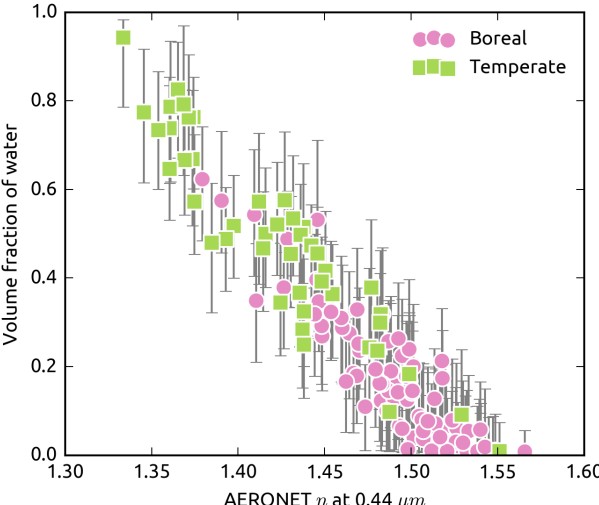

**Figure 4.** Inferred volume fraction of water. Error bars show interquartile range of inferred values resulting from uncertainties in AERONET particle properties.

### 3.3 Variability in particle properties

The identified AERONET observations of boreal and temperate smoke suggest distinctiveness in retrieved size distributions and refractive index (Fig. 3 (A–C)). The selected observations indicate that boreal emissions tend to have larger particles with median volume median radius value of 0.19 (0.17–0.21) compared to 0.17 (0.16–0.19) µm obtained for temperate smoke.

These differences may be influenced by differences in combustion phase between the biomes. Very intense and predominantly flaming fires emit larger particles than events with more important smouldering combustion (Reid et al., 2005a). Substantial differences exist comparing the complex index of refraction. Boreal plumes exhibit higher median $n$ value of 1.49 (1.47–1.52) in contrast to 1.43 (1.37–1.45) observed for plumes attributed to temperate forest fires. Although boreal smoke generally is more absorbing with median $k$ value 0.008 (0.007–0.01)$i$ compared to the 0.005 (0.004–0.008)$i$ value obtained for temperate emissions, plumes from both biomes are only weakly absorbing and characteristic $k$ values have a negligible influence on calculated $B_{ext}$. Variability in the real part of the refractive index between the plume categories, on the other hand, is larger and indicates differences in particle chemistry.

### 3.4 Inferred volume water fractions

Maxwell-Garnett medium approximation calculations using the discussed optical constants result in substantially different inferred water content for the two sources (4). The variability is mainly driven by the real part of the refractive index. Inferred median black carbon fractions are less than 1 % for both classes and thus have minimal impact on water content retrieval. Median water volume fraction for boreal fires is 0.15 (0.1–0.31), whereas temperate plumes have median value of 0.47 (0.29–





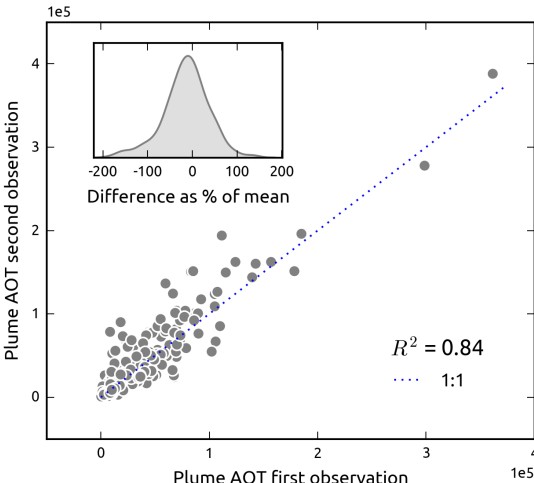

**Figure 5.** Fire-emitted AOT for 284 cases with two estimates for the same diurnal emission period starting and ending at 00:00 UTC, obtained at different stages of plume development. Inset shows distribution of the differences between the estimates as a percentage of their mean value.

$0.67)\,\%$. The derived values agree with water volume fractions inferred by Schuster et al. (2009) using a similar approach, although dust was not included as one of the components in our retrieval. The main limitations of the presented method are (i) the assumption that aerosols with $n \geq 1.53$ are dry and (ii) large uncertainties in the chosen $n$ values and different components used in the retrieval. In addition to increasing water content, formation of organic compounds may alter aerosol optical properties. Measured $n$ for dry ambient organic aerosol are typically lower than the 1.53 value used in this study, ranging from 1.47 to 1.53 (Dick et al., 2007) and appear to change with age (Rudich et al., 2007). Although the uncertainties in AERONET properties and particle density were propagated in the retrieval, water fractions inferred in this study critically depend on $n$ of the dry major component being close to 1.53. Any departures from this value result in inaccurate water uptake retrieval.

## 3.5 Simulated mass extinction efficiencies

The differences in plume particle properties, primarily $n$ and particle size, coupled with distinctiveness in inferred volume water fractions, drive differences in simulated $B_{ext}$ for the dry volume content of the plumes. Boreal plumes have larger particles, higher values of refractive index, but smaller water fractions and hence have lower median $B_{ext}$ value of 5.7 (5.1– 6.5), while emissions originating from temperate forests have a median $B_{ext}$ value of 6.7 (5.4–9.2) $\mathrm{m^2/g}$ due to inferred greater water content. The identified AERONET observations are for ambient plumes which are aged for at least 1 to 3 days, and consequently, computed $B_{ext}$ values for dry volume fractions are larger than the $4.7 \pm 0.7\,\mathrm{m^2/g}$ value suggested for dry aged boreal and temperate emissions (Reid et al., 2005b). Somewhat higher values ranging from 4.7 to 5.5 $\mathrm{m^2/g}$ were calculated




**Figure 6.** Daily estimated TPM from this study and GFED for individual fire events. Error bars represent difference between two TPM values for the days of emission for which two estimates were obtained. Shown are robust linear fits, $\beta$ parameter indicates the slope.

(Reid et al., 2005b) for a set of AERONET retrievals from North American boreal forest (Dubovik et al., 2002). The main difference between that aerosol climatology and the retrievals used in this study are in the real part of the refractive index. Dubovik et al. (2002) climatology for boreal smoke represents generally dryer plumes with an average $n$ value of 1.5, compared to 1.49 and 1.43 median $n$ values attributed to boreal and temperate emission in this study.

## 3.6 Interpretation of changes in smoke optical thickness

The increase in attributed AOT in aged plumes determined in this study is consistent with well documented smoke particle evolution. Aerosols grow considerably in size as plumes age. Particles undergo rapid changes during the first few hours after emission due to combined effects of condensation and coagulation (Reid and Hobbs, 1998), with reported growth rates in





volume median radius as high as $0.04\,\mu m$ per hour (Hobbs et al., 1996). On the time scales of days plume particles continue to grow in dense plumes but at substantially lower rates, primarily due to coagulation and hygroscopic growth. Reported increases in volume median radius at these time scales are in the order of $0.02$–$0.03\,\mu m$ (Reid et al., 2005a; Nikonovas et al., 2015). Condensation of organic and inorganic species and secondary particle production increase particle plume mass, while

coagulation only transforms particle distribution. Both processes alter smoke optical thickness mainly by enlarging scattering cross-section and scattering efficiency, which is a strong function of particle size. Condensation has been reported to increase particle mass by up to $30$ - $40\,\%$ in Amazonian plumes, but is thought to be important only during the first 24 hours at most (Reid and Hobbs, 1998). The inferred increase in fire-emitted AOT over the first two days of ageing reported in this current study only partially overlaps with this period. The first few hours of plume development when condensation is thought to be the

most active are not represented, therefore condensation is unlikely to contribute significantly towards the inferred AOT growth. A growth in volume median radius of $0.02\,\mu m$ due to coagulation theoretically could increase scattering efficiency by up to $30\,\%$ without changes in plume mass, but this process can not explain differences in the magnitude of AOT change observed between the biomes.

An additional factor driving changes in AOT is water uptake by smoke particles. Absorption of water depends on air relative

humidity and aerosol solubility which in turn tends to increase with atmospheric processing. It increases particle size further, enhancing scattering cross-section. Hygroscopic growth factors measured and inferred by optical methods for biomass burning smoke at $80\,\%$ range from 1.1 to more than 2 (Kotchenmther and Hobbs, 1998; Kreidenweis et al., 2001; Magi and Hobbs, 2003). Reid et al. (2005b) suggested an average enhancement factor of $1.35 \pm 0.2$. $B_{ext}$ values derived for dry volume fraction in this study suggest median scattering cross-section enhancement factors of 1.2 and 2 for boreal and temperate plumes, assuming

the 4.7 $B_{ext}$ value for dry smoke (Reid et al., 2005b).

Notably, the magnitude of AOT increase over time shown in figure 2 corresponds to inferred median water fractions for the two biomes. Temperate emissions exhibit generally hydrophilic particles with much greater water content, while boreal plumes seem to contain much less aerosol water. This distinctiveness could be due to different ratios of smouldering and flaming combustion. Field measurements indicate that prescribed burns and in particular wildfires in temperate regions have

lower combustion efficiencies (Urbanski, 2014). Temperate fires discussed in this study have lower mean FRP values and a less pronounced diurnal burning cycle, and the emitted plumes have higher ratios of night-time emissions. Smouldering night-time smoke has been reported to contain more soluble organic compounds (Hoffer et al., 2006), which could explain the presence of more hydrophilic aerosols in temperate plumes. In addition, factors not accounted for in this study, such as significant differences in relative humidity and atmospheric processing between the biomes, may be partly responsible for the inferred

variability in water uptake.

## 3.7  Daily TPM estimates for individual fires

On an individual event basis the relationships between daily particulate emissions given by the global inventories and this study exhibit varying degrees of agreement. Figure 6 shows this study's and GFED TPM for the events for which estimation was performed for at least seven diurnal cycles. Although some fires exhibit only fair or weak agreement, the result is nonetheless





encouraging considering error in AOT attribution and conversion to TPM method in this study, and large uncertainty associated with the daily burned area product (Giglio et al., 2013) on which GFED depends. Robust linear fits between GFED TPM and daily estimated TPM shown in figure 6 indicate considerable variability in slopes, even comparing the events with generally good agreement. This suggests distinctive combustion and emission characteristics for individual events. As well as variability on a per burning event basis, large differences exist when comparing relationships for fires in boreal and temperate forests. Notably, for every tonne of GFED TPM, this study shows TPM ranges from 0.46 to over 2 tonnes for boreal burning events, while for temperate fires the conversion factors range from approximately 1 to more than 5. The relationships are similar in terms of agreement comparing daily TPM estimates with other inventories (not shown), but scaling factors which are needed to reconcile the estimates differ.

## 3.8 Comparison of total emissions and emission coefficients

Total TPM emission estimates obtained in this study for the wildfires examined are large in comparison to FEER, and to a lesser degree GFED and GFAS inventories, but are smaller than QFED estimates (Fig. 7). QFED emissions are reported for $PM_{2.5}$ aerosol fraction only, which typically constitutes 70 to 85 % of TPM for the biomes discussed (Akagi et al., 2011). As a result, QFED TPM estimates should be approximately 20–40% higher than indicated in figure 7.

Substantial differences exist comparing the estimates for boreal and temperate fires. For boreal forest events, total TPM emissions for this study are in close agreement with the bottom-up GFED and GFAS TPM estimates. The agreement indicates that application of the proposed 2.2 enhancement factor (Kaiser et al., 2012) to GFAS TPM would overestimate boreal emissions for the events discussed. Regional AOT based QFED inventory suggests $PM_{2.5}$ emissions higher by 40%, while near-source FEER TPM estimates are smaller by a factor of 2.8 when compared to TPM for this study.

For temperate forests, a striking contrast exists between GFED and GFAS inventories and methods based on regional AOTs. The largest estimates are given by the QFED inventory, which suggests $PM_{2.5}$ emissions which are higher by 50% than the TPM estimates for this study. If bottom-up estimates of the boreal emissions agree well with this study's TPM, for temperate events the discrepancies are much larger. Scaling factors of 3.2 and 4 are needed to reconcile GFED and GFAS emissions with the estimates obtained in this study. FEER emissions are closer to bottom-up approaches suggesting much lower emitted TPM compared to the other top-down methods. This appears to be characteristic to North America as has been reported in Ichoku and Ellison (2014), indicating potential underestimation of the emissions in the region. For other continents, FEER generally predict higher TPM emissions than the bottom-up inventories and agree closely or even exceed QFED $PM_{2.5}$ estimates.

The above emission budgets suggest particulate matter emission coefficients of 27 (23–30) and 31 (24–37) $g$ per $MJ^{-1}$ of time integrated GFASv1.0 FRP (table 2). They comprise approximately 70 % of coefficients derived for QFED $PM_{2.5}$ emissions, and are 2.5 times larger than equivalent values derived using FEER emission coefficients. Notably, although differing in magnitude, all three top-down methods indicate slightly larger emission coefficients for temperate events. In contrast, more bottom-up approaches suggest 2.5 to 3 times larger emission coefficients for boreal forest. The TPM emissions factors employed in GFAS and GFED inventories are identical for both forest types, but large differences exist in consumed biomass




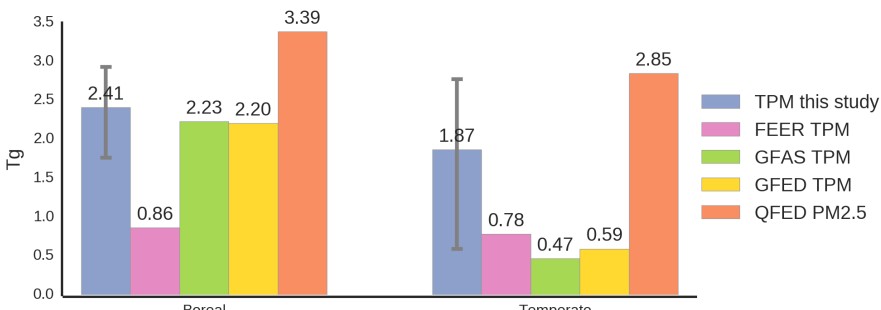

**Figure 7.** Total TPM emissions derived in this study and estimates for the same events and days of emission given by other methods. Error bars represent 95 % confidence interval determined taking into account uncertainties in (i) AERONET retrievals, (ii) inferred water fraction, (iii) particle density, (iv) modelled $B_{ext}$, and (v) estimated error in attributed daily AOT.

**Table 2.** Total particulate matter emission coefficients derived using GFASv1.0 FRP product and particulate matter emission estimates for the burning events discussed.

|  | Emission coefficients $(gMJ^{-1})$ | |
| --- | --- | --- |
|  | Boreal | Temperate |
| TPM this study | 27 (25–30) | 31 (24–37) |
| FEER TPM | 10 | 12 |
| GFAS TPM | 25 | 8 |
| GFED TPM | 25 | 10 |
| QFED PM$_{2.5}$ | 38 | 47 |

estimates. The GFAS inventory employs a three-fold larger FRP to dry matter combustion rate conversion factor for boreal fires, attributable to high organic soil content in the biome.

A number of factors may contribute towards the discrepancies between TPM estimates for this study and other methods. Relatively large estimates compared to near source GFED, GFAS and FEER inventories may be influenced by unaccounted
5 processes in ageing plumes. A several-fold growth in plume mass due to condensation and secondary particle production, however, seems implausible given that the reported magnitude of increase in particle mass driven by these processes is within 50 % (Reid and Hobbs, 1998). The difference may be partly due to large sizes of the events sampled in this study. Field measurements for large wildfires are scarce (Akagi et al., 2011; Urbanski, 2014), and such events are under-represented in compiled EFs. The agreement between top-down and bottom-up methods is better for boreal fires than it is for temperate events.
10 Fire events sizes are similar for both biomes, at least for the fires sampled. Therefore, it seems that fire size considerations alone





fail to explain the varying degree of agreement between this study's TPM, and GFED and GFAS estimates when comparing boreal and temperate cases. Comparably low FEER estimates, on the other hand, might be partly determined by sampled event size. Infrequent and large fires prevailing in North American forests make it difficult to reliably derive combustion coefficients from near the source imagery (Ichoku and Ellison, 2014).

Considering the above factors it seems that for the large fire events discussed, boreal emissions are underestimated by a factor close to 2 by FEER inventory. Temperate TPM appears to be underestimated by factors of 2 to 4 by FEER, GFED and GFAS. QFED on the other hand, seem to overestimate particulate emissions by 40 to 50 %. The previously suggested GFAS TPM 2.2 enhancement factor seems to represent an average value for the region. It is not required for boreal fires, and is close to 4 for temperate plumes. It is not clear if the underestimation by bottom-up GFED and GFAS is driven by low

emissions factors or biomass consumed estimates. More important smouldering combustion in temperate forests, however, suggests larger emission factors for this biome. Current measurements imply large underestimation for night-time emissions (Saide et al., 2015), suggesting the need for further investigation.

## 4    Conclusions

Refined particulate matter emission estimates are needed to improve future climate simulations and predict regional air quality

at shorter time scales. Existing global estimates differ by a factor of 2–4. The method presented in this study enables the estimation of daily TPM emissions from large wildfires with identifiable plumes and sufficient satellite AOT observations. Daily estimates take into account particulate matter emitted throughout a full diurnal cycle including both daytime and night-time emissions. Importantly, repetitive estimates are obtained for the same period of emission during up to three consecutive days of plume evolution allowing assessment of the AOT attribution error and systematic changes in smoke optical thickness

over time.

    Important insights are gained by partitioning plume AOT to daytime and night-time emissions. Night-time plume AOT seems to double when comparing observations of relatively young emissions of up to 18 h age to AOT attributed to the same period of emission from the following day's imagery. Only small changes are observed after the subsequent 24 hours of ageing. Daytime emitted AOT increases by approximately 30 % for temperate fires, but does not change over time in boreal smoke.

These changes have to be accounted for when reconciling emission estimates obtained near the source and from regionally dispersed aged plumes.

    We utilized available coinciding AERONET observations to infer characteristic aerosol water content in discussed plumes and parametrize Mie calculations of smoke mass extinction efficiency. Coinciding AERONET retrievals indicate median water volume fractions of 0.15 (0.1–0.31) and 0.47 (0.29–0.67) for boreal and temperate plumes respectively. Calculated $B_{ext}$ of the

dry particle fraction suggest median values of 5.7 (5.1–6.5) and 6.5 (5.5–9.2) $m^2/g$ for the two plume categories. The inferred water fractions indicate that hygroscopic growth accounts for the majority of the observed increase in plume optical thickness.

    Daily total particulate matter emissions determined using simulated $B_{ext}$ indicate differences in agreement with other inventories for the two forest type fires. For boreal fires which have higher median FRP values and burn predominantly during





the daytime, TPM estimates agree closely with GFED and GFAS inventories, are higher by a factor of 2 compared to FEER, and are lower by 30 % than QFED $PM_{2.5}$ estimates. For temperate events, which are characterised by small changes in active fire pixel count throughout the diurnal cycle and generally lower median FRP values, the discrepancies are larger. Our TPM estimates are lower than QFED $PM_{2.5}$ by 35 %, and higher by a factor of 4, 3.2 and 2.4 compared to GFAS, GFED and FEER

5   TPM estimates for the same emission events. The previously suggested scaling factor of 2.2 for GFAS particulate emissions is not required for boreal fires, but is too small for temperate events.

   The large fire event bias in this study and rapid ageing effects unaccounted for in this study could drive part of the difference, but are unlikely to explain all of it. Low FEER TPM for the discussed events could be attributed to these factors to a larger extent. The comparison of TPM obtained in this study to GFAS and GFED, however, suggest that TPM emission factors and

10  consumed biomass estimates are underestimated for temperate fires within the bottom-up datasets.





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
