# Peer review of "Particulate emissions from large North American wildfires estimated using a new top-down method"

_Atmospheric Chemistry and Physics, 2016_

## Referee Comment (RC1) · Anonymous Referee #1 · 31 Aug 2016

General Description of manuscript:

The authors use MODIS and AERONET AOD and HYSPLIT to estimate total particulate matter emissions from temperate and boreal fires and compare their estimates to values from biomass burning emission inventories.

General Comments:

Section 2.3 (Plume Dispersion Modelling) is not well described. Was the same particle number used for the two fire types (boreal and temperate)? Is this reasonable? Are there references to support this approach? What are the two different rates that are used for day and night emissions? Are there references to support different day/night

rates? When no fires were detected, why set the count to a minimum positive value instead of zero? Is this to account for undetected fires? What support is there for this approach? Please rewrite this section for clarity.

Water Content Retrieval. How do aerosol water fractions estimated in this work compare to aerosol water content that would be estimated using representative hygroscopic growth factors for representative relatively humidities?

The authors use "emission coefficients" in the text. Are these distinct from emission factors? If not, then rather use emission factors, as this is common terminology. If so, then please clarify the distinction in the text.

Specific Comments:

Abstract: FRP is used in the Abstract (and Introduction), but the acronym is only defined on page 10.

Abstract, line 20: Is "low bias" meant to be "negative bias"?

Page 2, line 19: Is the 3.4 correction factor applied to address an underestimate or overestimate in emissions? Please be specific.

Page 2, lines 32-34: This sentence is confusing. Please reword for clarity. Currently it reads that average EFs conceal the lack of spatial and temporal representativeness. Is this what the authors mean to say?

Page 3, lines 2-3: Isn't the approach in this study also susceptible to AOT retrieval errors and uncertainties in smoke particle properties?

Page 3, line 24: Please provide an appropriate reference or website for the MCD14ML data.

Page 4, lines 2-3: The approach used to identify fires is confusing. Are the "any pixels" pixels that include an active fire? Should there be consecutive active fire pixels within a 150 km radius? Please clarify in the text.

[Figure]

Page 5, lines 10-11: The change in resolution from nadir to the swath edges is true for the native resolution of the instrument, but the AOT product is at a nominal resolution of 10 km x 10 km.

Page 5, line 12: More appropriate is the MODIS AOT uncertainty for scenes with aerosols from boreal and temperate fires. Please either estimate the error by comparing MODIS and AERONET AOT or provide a value reported in the literature.

Figure 3: What is the mass concentration of individual aerosol components (inorganic, organic, black carbon) from boreal and temperate fires estimated in this study?

Page 6, lines 6-7: This sentence is unclear. Is this merely a scaling to convert AOT to fire-emitted particle number?

Table 1: Please change density to the Greek letter "rho" and enclose the units for density in square brackets for clarity.

Page 7, line 19: Why list both "organic carbon" and "organic matter"?

Page 8, lines 19-20: Incorrect in-text citation format for Ichoku and Ellison (2014). Please fix.

Page 10, lines 27-28: How different is median FRP for the two fire classifications if normalized to burned area or mass of biome burned?

Page 10, line 33: Would smouldering fires be detected as part of the large wildfire events that are isolated in this work?

Page 11, line 15: What is "(4)" referring to? Is this Figure 4?

Page 11, line 17: Are the median water volume values fraction or percent? Fraction is stated, but the "%" symbol is used (page 12, line 1).

Figure 5: This figure is out of sequence in the text. It is referred to in the text before Figure 4. Please fix.

[Figure]

Page 14, line 17: Is 80% relative humidity?

---

## Referee Comment (RC2) · Anonymous Referee #2 · 22 Dec 2016

Dear Authors, First thank you for a well-written description of a very large and hopefully significant experiment. Your approach includes numerous novel components, and represents a significant new attempt at reducing uncertainties in a crucial area where uncertainties remain very large, biomass burning emissions. My comments are organized roughly from important questions about the science to minor issues with the text, with typos and such at the end of this review. I look forward to seeing this study finalized and in print.

Your atmospheric simulation of smoke transport explicitly retains all smoke in the boundary layer. Wind shear in the vertical column, and other transport differences, will be a source of error in your estimates of smoke from many active fires that re-

lease smoke above the boundary layer (around 20% according to Val Martin ACP 2013 and Peterson 2014 JGR, but both of these estimates are based on satellite data with 1030am local overpass time, and thus likely conservative relative to overall fire behavior). Is there any way these effects can be estimated with the data you have corralled for this study?

The difference between boreal and temperate fires' day-night behavior is an interesting sidelight to this work. However, there is a good chance it is an artifact, and you must explore this before you finalize the paper. The basic idea is this: Terra and Aqua MODIS have nominal equatorial overpass times of 1030 and 1330 local solar time (LST, this can be calculated as UTC+[longitude/15.], where longitude is from -180 to 180), with the opposite orbital nodes crossing at 2230 and 0130 respectively. At higher latitudes, the wide MODIS swath covers a larger range of LST. Thus, a portion of the 2230 Terra swath will have LST<2100, and that portion will increase with latitude. So, if you define "daytime fires" as fires detected from 0900-2100LST, this will include all fires from the 1030 Terra overpass, all fires from the 1330 Aqua overpass, and depending on latitude, some fires from the 2230 Terra overpass. In order to avoid this, you should run the calculation using daytime=0600-1800LST, and see if the boreal-vs-temperate difference you observed holds up. I have attached a figure to illustrate this point, based on the MOD14 MODIS fire product.

page 4 Section 2.5 AOT attribution. This is the first of several very complicated steps, it is worth the effort to express very carefully how this was done. You have these ingredients:

1) modeled plume extent: this is a point cloud with the locations of all the smoke particle endpoints at solar noon on each day

2) MODIS AOT: you have the centroid location and retrieved AOT of each valid AOT retrieval on the day

3) background MODIS AOT: you have MODIS AOT and centroids for valid retrievals

from two days prior to construct the background estimate

As I understand it, you take these steps:

1) you interpolate modeled plume extent to 25km equal area grid, taking every grid cell that contains a portion of the plume and including it in the sample;

2) you interpolate MODIS AOD to the same 25km equal area grid;

3) you determine whether the number of valid same-day MODIS AOT data is at least (plume area /100km2)*0.8 ("80% coverage of plume area")

If #3:

4) You calculate the background AOT using the 2-days-prior AOT

5) you calculate the smoke AOT increment for each grid cell by subtracting background AOT from same-day AOT

If the median AOT increment is > 0:

6) you set negative AOT increments equal to zero

7) steps 1-6 are repeated for smoke transport times of up to 3 days, in increments of 12 hours.

8) If multiple days / multiple fires contribute to a grid cell AOT increment, you apportion the grid cell AOT increment to fire events and emission periods according to the number of smoke particles from the HYSPLIT simulation in each grid cell

Note that the cutoff in Step 3 will systematically eliminate coverage from scenes covered by the MODIS swath edge, because the smoke retrievals will be too few to cover the area based on the assumed 100km2 retrieval footprint.

page 2 Line 55: "Consumed biomass estimates inherit errors of... fire location..." The papers cited here cover a lot of ground, but I don't think they really cover errors associated with fire location. That source of uncertainty is described by Hyer and Reid

(GRL, 2007).

page 3 line 11 "larger than 100km2 and with duration longer than 7 days" Please elaborate slightly on the data and calculations used for these determinations, especially the 7 days.

page 3 line 15 the Stocks and Kasischke papers relate to fire size distribution in the boreal forest. While the dominance of large fires has been documented for certain parts of temperate north America (see Strauss, Bednar, and Mees, Forest Science, 1989), it does not hold for all areas and in any event is not covered by those citations.

Page 3 line 25 "particles were continuously injected" HYSPLIT in your configuration simulates transport of discrete particles, please specify the interval at which particles were released in HYSPLIT

Page 3 line 25 "vertically distributed" please specify the discrete intervals at which particles were released in HYSPLIT

page 3 line 25 "within the planetary boundary layer" as diagnosed by GDAS? Please specify.

page 4 line 6 "is about twice the size at swath edges." Actually, the single MODIS pixels increase roughly 8x in size from nadir to swath edge, and the 20x20 pixel footprints used by MxD04_L2 increase proportionally. However, there is significant overlap between MxD04_L2 footprints at swath edge, see Sayer et al. (http://www.atmos-meas-tech.net/8/5277/2015/). You may not need to quantitatively account for this for this study, but you should be aware of this.

Page 4 Section 2.4 how were AOT data selected from MxD04_L2 (quality flags, cloud fraction, etc.)?

Page 7 line 21: "the agreement between two or more estimates for the same emission period is reasonably static across the plume age categories." I do not see where this is shown in figures or tables—if it is there somewhere, please direct the reader to it when

you make this statement. One simple change would be to add a second panel to Figure 5 showing the agreement between Day 1 and Day 3 AOT for the same event/time pairs.

Minor corrections and typos

page 1 line 14 "take into account . . . efficiency when converting"

page 1 Line 27 "increase in AOT"

page 2 Line 50 "introduced by" => "results from"

page 2 Line 51 "within biome" => "within-biome"

page 2 Line 57 "are effected" => "are affected"

page 2 Line 63 "representative to" => "representative of"

page 2 Line 67 "need of" => "need for"

page 3 line 27 "proportional to active fire" => "proportional to the active fire"

page 3 line 35 "where detected" => "were detected"

page 3 line 36 "minimum positive value" => "minimum nonzero value"

page 3 line 37 capitalization

page 4 line 10 "have an upper limit"

page 4 line 83 "allow to retrieve" => "allow retrieval of"

page 4 line 75 "representative to" => "representative of"

Page 4 line 92 "(table 1)."

Page 10 line 90 biases

Page 11 line 19 overestimates

[Figure]

**Fig. 1.**

---

## Author Comment (AC1) · 2 Feb 2017

Response to anonymous reviewer's #1 comments.

The authors thank the reviewer for their constructive comments. Our responses to anonymous reviewer's #1 comments are detailed below. Reviewer's comments are in italics and our responses in standard font.

**General Comments:**

*Section 2.3 (Plume Dispersion Modelling) is not well described. 1) Was the same particle number used for the two fire types (boreal and temperate)? Is this reasonable? Are there references to support this approach? 2) What are the two different rates that are used for day and night emissions? Are there references to support different day/night rates? 3) When no fires were detected, why set the count to a minimum positive value instead of zero? Is this to account for undetected fires? What support is there for this approach? Please rewrite this section for clarity.*

The authors apologise for the lack of clarity in the section. The section has been rewritten clarifying the steps undertaken to perform dispersion simulations, and addressing the specific questions raised in this comment. The answers (numbered in the reviewers comment) are listed below.

1) HYSPLIT particle number emitted per hour per fire detection was the same for both biomes. This was done for several reasons. Firstly, total particle number is not directly linked to particulate emission estimates. For example, if a grid cell has AOT value of 1, and 100 HYSPLIT particles are located within the cell during the satellite overpass, 80 of which were emitted two diurnal cycles ago, and 20 during the previous diurnal cycle, the grid cells AOT is split accordingly between the emission periods. Equivalently, if there are any particles emitted from different fire events, grid cells AOT is divided both between different emission periods and different fire events. The estimates for the two periods would not be different if there were more or less particles within the cell, its the relative ratios which matter. Secondly, the aim of the manuscript was to provide independent estimates therefore we tried to avoid using existing emission factors or emission coefficients (top-down emission estimates per unit of fire radiative energy). Notably, even if emission rates are indeed different for boreal and temperate events, the assumed identical particle numbers in the analysis would not have influenced the results significantly. This is due to the fact that mixing of the plumes from different biomes was minimal.
2) The emitted particle number per hour and fire detection were identical for daytime and nigh-time periods.
3) The count was set to minimum non-zero value to avoid total shutdown of emissions for a time period for which no MODIS fire detections were obtained, most likely due to cloud cover. This is an unlikely scenario for the long duration burning episodes presented in the manuscript.

The updated section:

Smoke transport for the selected fire events was simulated with the HYSPLIT model (Draxler et al., 2003). Plume dispersion from a source location was represented by the motion of a large number of discrete particles moved by the wind field with mean and random components. Global Data Assimilation System (GDAS) meteorological archive data was employed to drive the model.

For each day of burning, particles were continuously released into the model domain from the locations of the individual active fire detections within the fire event. In order to represent fire diurnal cycle, different MODIS active fire observations were used to release particles for two 12 hour intervals representing day and night emissions 09:00 to 21:00 and 21:00 to 09:00 local time respectively. Emission source number and locations for daytime periods were determined from the highest number of fire detections observed during a single either Terra of Aqua daytime

overpass with 10.30 and 13.30 equatorial crossing time. Similarly, emitted particle source numbers for the night periods were determined by the largest burning extent observed during one of the night-time overpasses with 22.30 and 1.30 equatorial crossing times.  Notably, the Terra overpass at 22.30 in high latitudes makes observations of regions where local time is earlier than 21:00. In this study, however, all fires detected during this overpass were classed as night-time observations. If no valid observations were available for some of the time intervals, the count and fire pixel locations were set to a~minimum non-zero value estimated for the burning episode from all daytime or night-time observations. This was done to avoid total temporary shut-down of the emissions, which is an unlikely scenario for a long duration burning episodes. Every hour, 20 particles were released for each fire pixel. As a result, emitted particle number for a burning episode was determined by the number of active fire pixels observed during a given time period.

Particles were uniformly distributed between the surface and the top altitude of the planetary boundary layer as given in GDAS archive. Satellite based plume height estimates (Val Martin et al., 2010, Peterson et al., 2014) indicate that in up to 80% of the events analysed, injection heights were limited to the planetary boundary layer.  While confinement of the emissions to the mixing layer underestimates injection height for the most energetic burning episodes, such configuration should nonetheless represent the majority of burning episodes.

Throughout the simulations, modelled particle positions, their age and source burning event identifier were recorded each day at local solar noon. The generated point clouds were later used to compare against Terra and Aqua Aerosol Optical Thickness (AOT) observations. "

*Water Content Retrieval. How do aerosol water fractions estimated in this work compare to aerosol water content that would be estimated using representative hygroscopic growth factors for representative relatively humidities?*

The inferred median water volume fractions would equate to geometric hygroscopic growth factors (gHGF) of 1.05 and 1.24 for boreal and temperate plumes respectively. Such factors suggest that boreal plumes belong to "nearly-hydrophobic'' group (gHGF 1.0-1.11) while temperate plumes fall into ''less-hygroscopic'' category (gHGF range 1.11–1.33) as suggested in Swietlicky et al., 2008 review of measured growth factors. Note that the measured growth factors were recorded at 90% relative humidity, while water fractions inferred in our study were obtained at ambient relative humidities. In result, a direct comparison is not very meaningful, but our numbers do seem to fit reasonably well. Smoke aerosols are most often classified as less-hygroscopic with gHGFs of 1.11-1.33.  We have added a short discussion:

"These estimates compare favourably to measured factors for biomass burning smoke (Swietlicky et al., 2008), indicating nearly-hydrophobic particles for boreal plumes, while temperate plumes could be classed as less-hygroscopic. Notably, measured geometric hygroscopic growth factors are reported at 90% relative humidity. In contrast, water volume fractions inferred
in this study are representative of ambient humidity levels, and as a result direct comparison
is not very meaningful.

*The authors use "emission coefficients" in the text. Are these distinct from emission factors? If not, then rather use emission factors, as this is common terminology. If so, then please clarify the distinction in the text.*

The term "emission coefficients" is used to contrast the top-down particulate emission estimates per unit of FRP with measured emission factors which are obtained for unit of fuel burned. Such nomenclature was used in other top-down studies (Kaiser et al., 2012; Ichoku and Elison 2014). We have rephrased the paragraph introducing the concept:

"A~top-down global gridded Fire Energetics and Emissions Research (FEERv1) (Ichoku et al., 2014)  product is based on collocated satellite FRP and AOT observations. Inferred total particular matter emissions rates are linked to observed FRP. The estimated TPM emission coefficients allow direct conversion from time integrated FRP to emitted particulate matter without invoking the emissions factors."

***Specific Comments:***

*Abstract: FRP is used in the Abstract (and Introduction), but the acronym is only defined on page 10.*

We have replaced the acronym in abstract with "fire radiative power", and added the definition to the first instance in the text.

*Abstract, line 20: Is "low bias" meant to be "negative bias"?*

Indeed, "negative bias" was the intended wording, changed accordingly.

*Page 2, line 19: Is the 3.4 correction factor applied to address an underestimate or overestimate in emissions? Please be specific.*

The correction factor was needed to address underestimation in emissions. The ambiguous statement has been changed to "enhancement factor".

*Page 2, lines 32-34: This sentence is confusing. Please reword for clarity. Currently it reads that average EFs conceal the lack of spatial and temporal representativeness. Is this what the authors mean to say?*

The sentence has been rephrased to "Average EFs for different biomes are based on small sample numbers for some areas, and conceal large variability in individual measurements…"

*Page 3, lines 2-3: Isn't the approach in this study also susceptible to AOT retrieval errors and uncertainties in smoke particle properties?*

We perhaps didn't make this clear enough, but the statement was "top-down aerosol inversions are affected by AOT retrieval error and large uncertainties in assumed smoke particle properties". The approach taken by this study is "top-down" by definition, therefore the statement was directed at our and other similar studies. To make it clear that it applies to any top-down method including this study, we have replaced "top-down aerosol inversions" with "top-down approaches".

*"Page 3, line 24: Please provide an appropriate reference or website for the MCD14ML data.*

*Reference (Giglio et al., 2006) has been added.*

*Page 4, lines 2-3: The approach used to identify fires is confusing. Are the "any pixels" pixels that include an active fire? Should there be consecutive active fire pixels within a 150 km radius? Please clarify in the text.*

"any pixels" was used meaning any MODIS fire detections. This whole section has been rewritten for clarity as been requested by the other reviewer:

Large and long-lived fire events, likely strong emission sources, were identified and selected for the analysis. Burning episodes larger than 100km^2 are not numerous, but account for more than 80%

of total burned area in boreal North America (Stocks 2002, Kasischke 2002), and are a dominant mode of burning in parts of temperate regions as well (Strauss et al., 1989). In order to identify such events, individual MODIS active fire detections were agglomerated into large wildfire events by performing two step spatial-temporal clustering. First, any MODIS fire detections located closer than 10km in space and 24 hours in time were grouped together. Single detections not assigned to any of the formed clusters were removed from further analysis. The clusters were then filtered by selecting events with (i) spatial bounding box containing all fire detections belonging to the cluster larger than 100km^2 and (ii) duration longer than 7 days. The duration was determined by the time span between the first and the last MODIS active fire detection belonging to the cluster. The burning was considered uninterrupted if the largest temporal interval between subsequent MODIS fire observations was less than 24 hours. During the second step of clustering, any of the selected events active at the same time and located closer than 150km were grouped into large burning episodes, assigning a unique source label. These events were classified into boreal and temperate fires using the dominant emission source given in the GFEDv4 inventory for areas and periods when the events were active.

*Page 5, lines 10-11: The change in resolution from nadir to the swath edges is true for the native resolution of the instrument, but the AOT product is at a nominal resolution of 10 km x 10 km.*

It is our understanding that Collection 5 MODIS dark target algorithm uses blocks of 20 x 20 500m MODIS pixels while deep blue algorithm employs block 10 x 10 of 1 km MODIS pixels (Levy et al., 2013; Sayer et al., 2015). The block size is fixed across the swath, and hence the footprint increases proportionally with individual pixel size. The original statement in the manuscript "pixel size increases twice at the edge of the swath" was indeed incorrect as pointed out by the other reviewer. In fact, AOT pixel footprint increases approximately 9 times.

*Page 5, line 12: More appropriate is the MODIS AOT uncertainty for scenes with aerosols from boreal and temperate fires. Please either estimate the error by comparing MODIS and AERONET AOT or provide a value reported in the literature.*

We have added the regional analysis of Collection 5 MODIS AOT retrieval uncertainties (Hyer et al., 2011) to the paragraph:

"A regional MODIS M{*}D04_L2 AOT product validation (Hyer et al., 2011) indicates that performance varies greatly within North America. The study found that for 0.2<AOT<1.4 conditions, root mean squared error varies from 0.01+0.51xAOT in arid Western America where retrieval is hindered by bright surfaces, to 0.01+0.31xAOT in boreal forests and 0.3+0.12xAOT in Eastern USA. The study reported positive bias in MODIS AOT for some locations, in particular for retrievals at extremely high aerosol loadings."

*Figure 3: What is the mass concentration of individual aerosol components (inorganic, organic, black carbon) from boreal and temperate fires estimated in this study?*

While we did estimate volume fractions of black carbon and a mixture of organic and inorganic compounds represented by n values close to 1.53 when retrieving water content, a decision was taken not to present them in this manuscript. We believe that the discussion on different components should be left out of this manuscript which is focused on TPM estimates. In any case, the retrieved median volume fractions of black carbon were relatively low and hardly different when comparing the two categories, 0.01 and 0.009 for boreal and temperate plume observations respectively. Notably, we did not retrieve absorbing organic carbon or "brown carbon" fractions by utilising its wavelength dependent absorption. Organic carbon was represented as part of the third component encompassing organic and inorganic compounds.

*Page 6, lines 6-7: This sentence is unclear. Is this merely a scaling to convert AOT to fire-emitted particle number?*

The section has been updated clarifying the steps used when attributing AOT. Bellow is the relevant paragraph:

"If a mixture of particles was found within a cell indicating that multiple fires and multiple emission periods contributed towards the grid cell AOT, the attribution was performed by apportioning a~grid cell's fire-emitted AOT in proportion to the numbers of modelled particles released during the emission periods and with origin found within the grid cell. For example, if a grid cell had AOT value of 1, and 100 HYSPLIT particles were located within the cell during the satellite overpass, 80 of which were emitted two diurnal cycles ago, and 20 during the previous diurnal cycle, the grid cell AOT was split accordingly between the emission periods. Panels K and L in figure 1 illustrate partitioning of total plume AOT to two different emission periods. Similarly, if there were any particles emitted from different fire events, grid cells AOT was divided both between different emission periods and different fire events."

*Table 1: Please change density to the Greek letter "rho" and enclose the units for density in square brackets for clarity.*

Thank you for noting this, the symbol was changed accordingly.

*Page 7, line 19: Why list both "organic carbon" and "organic matter"?*

Thank you for pointing this out. The redundant phrase has been removed.

*Page 8, lines 19-20: Incorrect in-text citation format for Ichoku and Ellison (2014).
Please fix.*

The citation has been fixed.

*Page 10, lines 27-28: How different is median FRP for the two fire classifications if normalized to burned area or mass of biome burned?*

We do agree that such a comparison would have been interesting. However, we did not employ area burned datasets in the study, and did not perform a comparison with biomass burned estimates. As a result, it is not feasible to include this information at this stage. In any case, the comparison of per-pixel FRP values is only a sideline to this study, and we did not place much weight on it, but felt the need to report it.

*Page 10, line 33: Would smouldering fires be detected as part of the large wildfire events that are isolated in this work?*

The authors are not aware of a method to directly detect smouldering combustion by remote sensing means. Median FRP values alone certainly do not provide enough information. However, smouldering combustion has been reported to be more important during night (Reid et al 2005); and that smoke from smouldering combustion can be lofted and entrained  into main plumes by convection (Urbanski 2013). The statement is therefore merely an interpretation based on literature, not something which can be confirmed by the data employed.
To avoid any possible confusion, We have removed the statement linking lower FRP and night time burning to smouldering combustion.

*Page 11, line 15: What is "(4)" referring to? Is this Figure 4?*

The reference to figure 4 has been corrected.

*Page 11, line 17: Are the median water volume values fraction or percent? Fraction is stated, but the "%" symbol is used (page 12, line 1).*

We apologise for this error. The median volume fractions were reported, not percentages. The symbol has been removed.

*Figure 5: This figure is out of sequence in the text. It is referred to in the text before Figure 4. Please fix*

Figure reference order has been corrected.

*Page 14, line 17: Is 80% relative humidity?*

Yes, relative humidity was meant. Added to the text.

---

## Author Comment (AC2) · 2 Feb 2017

**Response to anonymous reviewer's #2 comments.**

The authors thank the reviewer for their insightful and constructive comments. Our responses to anonymous reviewer's #2 comments are detailed below. Reviewer's comments are in italics and our responses in standard font.

**General Comments:**

*Your atmospheric simulation of smoke transport explicitly retains all smoke in the boundary layer. Wind shear in the vertical column, and other transport differences, will be a source of error in your estimates of smoke from many active fires that release smoke above the boundary layer (around 20% according to Val Martin ACP 2013 and Peterson 2014 JGR, but both of these estimates are based on satellite data with 1030am local overpass time, and thus likely conservative relative to overall fire behaviour). Is there any way these effects can be estimated with the data you have corralled for this study?*

Indeed, restricting injection heights to the top of the planetary boundary layer is a limitation to the method. While the quantification of error and bias introduced by this limitation was not achieved, the authors expect that this effect is small when compared to other sources of uncertainty, both accounted and unaccounted for in the manuscript. Energetic burning episodes when smoke is injected directly into the free-troposphere can be expected to have significantly different transport pattern when compared to our within-PBL transport model output. And as a result, many of such cases should have been filtered out by the MODIS AOT and modelled plume extent matching step in the analysis. Consequently, this limitation should be primarily manifested as a selection bias in the results, excluding the most energetic events from the sample.

*The difference between boreal and temperate fires' day-night behaviour is an interesting sidelight to this work. However, there is a good chance it is an artifact, and you must explore this before you finalize the paper. The basic idea is this: Terra and Aqua MODIS have nominal equatorial overpass times of 1030 and 1330 local solar time (LST, this can be calculated as UTC+[longitude/15.], where longitude is from -180 to 180), with the opposite orbital nodes crossing at 2230 and 0130 respectively. At higher latitudes, the wide MODIS swath covers a larger range of LST. Thus, a portion of the 2230 Terra swath will have LST<2100, and that portion will increase with latitude. So, if you define "daytime fires" as fires detected from 0900-2100LST, this will include all fires from the 1030 Terra overpass, all fires from the 1330 Aqua overpass, and depending on latitude, some fires from the 2230 Terra overpass. In order to avoid this, you should run the calculation using daytime=0600-1800LST, and see if the boreal-vs-temperate difference you observed holds up. I have attached a figure to illustrate this point, based on the MOD14 MODIS fire product.*

The authors want thank the reviewer for this detailed comment. The problem here was that we did not state clearly how the daytime to night-time fire pixel counts were derived. The 0900-2100LST periods were used to emit particles, not to determine if an observation represents daytime of night-time burning. The authors were aware of the large spread of LST values for northern latitudes and therefore all fire detections from Terra overpasses with equatorial crossing time 2230 were considered to be night-time observations.

Please see the figure bellow, which shows counts of fire detections per local hour and day - night classification for all fire events analysed in the manuscript.

[Figure]

We have added a clarification on how fire detections were classified as daytime or night-time fires into section 2.33

"Emission source number and locations for daytime periods were determined from the highest number of fire detections observed during a single either Terra of Aqua daytime overpass with 10.30 and 13.30 equatorial crossing time. Similarly, emitted particle source number for the night periods were determined by the largest burning extent observed during one of the night-time overpasses with 22.30 and 1.30 equatorial crossing times. Notably, the Terra overpass at 22.30 in high latitudes makes observations of regions where local time is earlier than 21:00. In this study, however, all fires detected during this overpass were classed as night-time observations."

*page 4 Section 2.5 AOT attribution. This is the first of several very complicated steps, it is worth the effort to express very carefully how this was done. You have these ingredients: 1) modelled plume extent: this is a point cloud with the locations of all the smoke particle endpoints at solar noon on each day 2) MODIS AOT: you have the centroid location and retrieved AOT of each valid AOT retrieval on the day 3) background MODIS AOT: you have MODIS AOT and centroids for valid retrievals from two days prior to construct the background estimate As I understand it, you take these steps: 1) you interpolate modeled plume extent to 25km equal area grid, taking every grid cell that contains a portion of the plume and including it in the sample; 2) you interpolate MODIS AOD to the same 25km equal area grid; 3) you determine whether the number of valid same-day MODIS AOT data is at least (plume area /100km2)\*0.8 ("80% coverage of plume area")*

*If #3:*
*4) You calculate the background AOT using the 2-days-prior AOT 5) you calculate the smoke AOT increment for each grid cell by subtracting background AOT from same-day AOT If the median AOT increment is > 0: 6) you set negative AOT increments equal to zero 7) steps 1-6 are repeated for smoke transport times of up to 3 days, in increments of 12 hours. 8) If multiple days / multiple fires contribute to a grid cell AOT increment, you apportion the grid cell AOT increment to fire events and emission periods according to the number of smoke particles from the HYSPLIT simulation in each grid cell Note that the cutoff in Step 3 will systematically eliminate coverage from scenes covered by the MODIS swath edge, because the smoke retrievals will be too few to cover the area based on the assumed 100km2 retrieval footprint.*

Thank you for the detailed suggestions and advice on how to describe the method. This section is conveying very complex processing steps and we perhaps didn't achieve sufficient clarity. The section has been rewritten following the above advice and clarifying what was not stated properly. It

is our belief that the method description now reads better and it is much more clear what was done in order to obtain our estimates:

"Elevated MODIS AOT observations were attributed to a specific fire event and emission period by comparing above background MODIS AOT retrievals to plume extent modelled by HYSPLIT (Fig. 1). The attribution required to determine three pieces of information; (i) event-specific background AOT value, (ii) modelled plume extent at local solar noon for each day of burning and (iii) coinciding MODIS AOT observations. First of all, background AOT value was estimated for each of the selected burning events. It was determined by the median value of the AOT retrievals within 150 km radios from the fire event centroid observed two days prior to ignition. For each day of fire activity, modelled plume extent (Fig. 1 (D–F)) was determined from the locations of all HYSPLIT particle endpoints at solar noon, and AOT observations (Fig. 1 (A–C)) from either Terra or Aqua platform with the highest spacial coverage for the day and plume area were selected.

After the required information was obtained, the following steps were performed for each day of burning attempting to estimate fire-emitted AOT. First, plume regions bounding the particles released during the previous three daytime and night-time emission periods were identified. Estimation of emission was attempted individually for each of the regions representing plume areas emitted during a specific time interval. This allowed the estimation of emitted AOT for up to three previous days from a single day of MODIS imagery. Importantly, such approach allows the estimation for some emission periods even if full MODIS plume overview is not available. Emitted AOT attribution was performed for the plume regions which satisfied two conditions. The region had (i) at least 80 % of MODIS AOT areal coverage assuming that a single AOT pixel represents 100 km 2 area, and (ii) with-region AOT median value was higher than the estimated background value for the fire event.

MODIS AOTs for the selected plume regions were interpolated to a 25 km resolution equal area grid (Fig. 1 (G--I)) by employing radial basis function interpolation with a~linear kernel. Fire-emitted AOT were estimated by subtracting the background value from the within-plume AOT.  The estimated fire-emitted AOT in every within-plume grid cell was apportioned to different emission periods and different sources based on information on release time and source of the HYSPLIT particles contained within the cell. If all particles found within a grid cell were released during the same emission period and originated from a single source, the cell's AOT was simply attributed to that emission period and source. If a mixture of particles were found within a cell, indicating that multiple fires and multiple emission periods contributed towards the grid cell AOT, the attribution was performed by apportioning a~grid cell's fire-emitted AOT in proportion to the numbers of modelled particles released during the emission periods and with origin found within the grid cell. For example, if a grid cell had AOT value of 1, and 100 HYSPLIT particles were located within the cell during the satellite overpass, 80 of which were emitted two diurnal cycles ago, and 20 during the previous diurnal cycle, the grid cell AOT was split accordingly between the emission periods. Panels K and L in figure 1 illustrate partitioning of total plume AOT to two different emission periods. Similarly, if there were any particles emitted from different fire events, grid cell AOT was divided both between different emission periods and different fire events."

*page 2 Line 55: "Consumed biomass estimates inherit errors of fire location". The papers cited here cover a lot of ground, but I don't think they really cover errors associated with fire location. That source of uncertainty is described by Hyer and Reid (GRL, 2007).*

Thank you for this suggestion, the reference has been added to support the relevant statement.

*page 3 line 11 "larger than 100km2 and with duration longer than 7 days" Please elaborate slightly on the data and calculations used for these determinations, especially the 7 days.*

The paragraph has been updated detailing what was meant by "larger than 100km2" and "duration longer than 7 days". Fire event size was determined by the size of the bounding box containing all fire detections for the event. Event duration was determined by the time span between the first and the last MODIS fire detection for the event. Burning episode was considered continuous if there were no 24h or longer gaps between the consecutive observations.

*page 3 line 15 the Stocks and Kasischke papers relate to fire size distribution in the boreal forest. While the dominance of large fires has been documented for certain parts of temperate north America (see Strauss, Bednar, and Mees, Forest Science, 1989), it does not hold for all areas and in any event is not covered by those citations.*

Thank you for noting this. The paragraph was changed clarifying: "Burning episodes larger than 100km^2 are not numerous, but account for more than 80% of total burned area in boreal North America (Stocks 2002, Kasischke 2002)" and "are a dominant mode of burning in parts of temperate regions as well (Strauss et al., 1989)

*Page 3 line 25 "particles were continuously injected" HYSPLIT in your configuration simulates transport of discrete particles, please specify the interval at which particles were released in HYSPLIT Page 3 line 25 "vertically distributed" please specify the discrete intervals at which*
*particles were released in HYSPLIT page 3 line 25 "within the planetary boundary layer" as diagnosed by GDAS? Please specify.*

This section has been rewritten as requested by other reviewer, addressing all issues raised here as well. The specific points are clarified bellow.

"particles were continuously injected" has been changed to "20 particles released per hour per each active fire pixel within the fire event".
"vertically distributed" was changed to "uniformly distributed between the surface and the top of the boundary layer as given in the GDAS archive"

*page 4 line 6 "is about twice the size at swath edges." Actually, the single MODIS pixels increase roughly 8x in size from nadir to swath edge, and the 20x20 pixel footprints used by MxD04_L2 increase proportionally. However, there is significant overlap between MxD04_L2 footprints at swath edge, see Sayer et al. (http://www.atmos-meastech.net/8/5277/2015/). You may not need to quantitatively account for this for this study, but you should be aware of this.*

Thank you for noting this discrepancy. MODIS pixel size is indeed ~9 times larger at the edge of the swath as demonstrated in the suggested study (Sayer et al., 2015). The relevant paragraph has been updated stating "...10 x 10 spatial resolution at nadir. MODIS pixel size increases with view angle, and pixels at the edge of the swath are approximately 9 times larger.

*Page 4 Section 2.4 how were AOT data selected from MxD04_L2 (quality flags, cloud fraction, etc.)?*

The selection criteria were added to the text "all retrievals with quality assurance confidence > 0 were selected. To maximise coverage, no cloud fraction filtering was applied."

*Page 7 line 21: "the agreement between two or more estimates for the same emission period is reasonably static across the plume age categories." I do not see where this is shown in figures or tables. If it is there somewhere, please direct the reader to it when you make this statement. One simple change would be to add a second panel to Figure 5 showing the agreement between Day 1 and Day 3 AOT for the same event/time pairs.*

*Indeed this statement was not supported by any of the figures. As suggested, we have now included panel B to Figure 5 showing difference between two estimates obtained at different stages of plume development. The statement now refers to the new figure.*

[Figure]

*Minor corrections and typos*

Thank you for taking time to find these mistakes. The authors apologise for leaving them in. All of these now have been corrected.